# A clinical diabetes risk prediction model for prediabetic women with prior gestational diabetes

**Bernice Man**[1]*, **Alan Schwartz**[2], **Oksana Pugach**[3], **Yinglin Xia**[4], **Ben Gerber**[1]

**1** Division of Academic Internal Medicine and Geriatrics, Department of Medicine, University of Illinois at Chicago, Chicago, Illinois, United States of America, **2** Department of Medical Education, Department of Pediatrics, University of Illinois at Chicago, Chicago, Illinois, United States of America, **3** Institute for Health and Research Policy, University of Illinois at Chicago, Chicago, Illinois, United States of America, **4** Division of Gastroenterology and Hepatology, Department of Medicine, University of Illinois at Chicago, Chicago, Illinois, United States of America

* bernicem@uic.edu

**Data Availability Statement:** This secondary analysis used publicly accessible data from the Diabetes Prevention Program which is maintained by the National Institute of Diabetes and Digestive and Kidney Diseases (NIDDK) Central Repository

## Abstract

### Introduction

Without treatment, prediabetic women with a history of gestational diabetes mellitus (GDM) are at greater risk for developing type 2 diabetes compared with women without a history of GDM. Both intensive lifestyle intervention and metformin can reduce risk. To predict risk and treatment response, we developed a risk prediction model specifically for women with prior GDM.

### Methods

The Diabetes Prevention Program was a randomized controlled trial to evaluate the effectiveness of intensive lifestyle intervention, metformin (850mg twice daily), and placebo in preventing diabetes. Data from the Diabetes Prevention Program (DPP) was used to conduct a secondary analysis to evaluate 11 baseline clinical variables of 317 women with prediabetes and a self-reported history of GDM to develop a 3-year diabetes risk prediction model using Cox proportional hazards regression. Reduced models were explored and compared with the main model.

### Results

Within three years, 82 (25.9%) women developed diabetes. In our parsimonious model using 4 of 11 clinical variables, higher fasting glucose and hemoglobin A1C were each associated with greater risk for diabetes (each hazard ratio approximately 1.4), and there was an interaction between treatment arm and BMI suggesting that metformin was more effective relative to no treatment for BMI $\geq$ 35kg/m$^2$ than BMI < 30kg/m$^2$. The model had fair discrimination (bias corrected C index = 0.68) and was not significantly different from our main model using 11 clinical variables. The estimated incidence of diabetes without treatment was 37.4%, compared to 20.0% with intensive lifestyle intervention or metformin treatment for women with a prior GDM.

https://repository.niddk.nih.gov/studies/dpp/. The NIDDK Central Repository collects and maintains data from designated NIDDK studies to facilitate new analyses after a study's original data coordinating center closes. Our analysis used only publicly released data, without any special access privileges from the NIDDK central repository. Formed data (S03, S05 and q03) and non-formed data (basedata, lab and events) were used for our analysis. The Requests can be made by registration through https://repository.niddk.nih.gov/user/login/?next=/requests/data-request/. Data requests must be accompanied by completion of a data agreement form which includes a brief description of the research, research objective and design, analysis plan, and statement of public use.

**Funding:** B.M. is funded by the UIC Center for Research on Women and Gender, UIC College of Medicine and UIC Department of Medicine and Division of Academic Internal Medicine & Geriatrics.

**Competing interests:** The authors have declared that no competing interests exist.

## Conclusions

A clinical prediction model was developed for individualized decision making for prediabetes treatment in women with prior GDM.

## Introduction

Gestational diabetes mellitus (GDM) is a common medical complication of pregnancy with a prevalence of 7–9% [1–3]. Although GDM typically resolves after delivery, women remain at high risk of developing type 2 diabetes mellitus, with a cumulative incidence of 30–50% within 5–10 years of the index pregnancy [3, 4]. Women with a history of GDM are more likely to progress to diabetes compared to those without GDM despite the same degree of impaired glucose tolerance at baseline [5]. Thus, the American Diabetes Association (ADA) and American College of Obstetricians and Gynecologists recommend diabetes screening at 4–12 weeks postpartum and every 1–3 years thereafter [6, 7]. For those with prediabetes (with early evidence of abnormal glycemic parameters), screening is recommended annually.

For most adults with prediabetes, treatment involves intensive lifestyle intervention (ILI) and/or metformin. However, the expected treatment response differs based on individual clinical factors (e.g., beta cell function), including history of GDM. Specifically, among those with prior GDM, risk reduction with metformin is comparable to highly-effective ILI (both reducing 3-year risk of progression to diabetes by approximately 50%) [5]. In contrast, metformin is less effective than ILI in the general population, including parous women without a history of GDM [8]. Overall, evidence suggests there is heterogeneity in prediabetes treatment response when GDM history is considered [5, 9].

Individualized risk prediction with estimated treatment response may inform prediabetes treatment decisions [10, 11]. Women with prior GDM can become aware of their future risk of diabetes and potential benefit from metformin and/or ILI. Individualized risk/benefit assessment may improve diabetes risk perception and prediabetes treatment decisions in this high-risk population. Such an assessment may optimize the appropriate use of metformin and ILI, considering the risks and costs, personal preferences, and potential benefits. Individual risk estimation may improve clinical decision making and make diabetes prevention efforts more efficient, cost-effective and patient-centered [10, 11].

There are numerous models available to predict the risk of developing diabetes for the general population [10–15]. However, few use multivariable models to facilitate tailoring preventive interventions to individuals [10, 11, 16, 17]. Of the models specifically developed for women with prior GDM, predictors commonly include measures obtained during or soon after pregnancy (e.g., insulin use during pregnancy or breastfeeding history) [15]. In a multivariable analysis of 174 women with GDM in Sweden, predictors of diabetes within 5 years postpartum include parity, a first-degree family member with diabetes, fasting glucose and hemoglobin A1c (HbA1c) levels during pregnancy [18]. In another GDM-specific model, non-European origin, glucose concentration from the 75 gram, 2-hour oral glucose tolerance test (OGTT) at pregnancy, and body mass index (BMI) at 1–2 years post-partum were predictive of diabetes [19]. Use of models incorporating peripartum measures may be limited because such measures may not be consistently obtained and are not often readily available to different clinicians providing care years later. Additionally, these models do not consider prediabetes treatment response, and may not be generalizable to diverse ethnic minority populations in the U.S. or to women who have already developed prediabetes after pregnancy [18, 19].

Our objective was to develop a clinical diabetes risk prediction model specific for women with prior GDM. The model includes estimation of treatment response to metformin, ILI, or neither (placebo) and can be incorporated into a decision aid for clinicians to use in prediabetes treatment counseling.

## Materials and methods

### Study design

We conducted a secondary analysis of the Diabetes Prevention Program (DPP) data [8]. The analysis was conducted using the publicly-available data which included no participant identifying information. Our study findings can be replicated in its entirety from the DPP data available only upon request to the National Institute of Diabetes and Digestive and Kidney Diseases Central Repositories (https://repository.niddk.nih.gov/studies/dpp/). The DPP was a randomized, controlled clinical trial comparing the effectiveness of ILI, metformin (850 mg twice daily), and placebo for diabetes prevention over a mean follow-up period of 3.2 years. In addition, standard lifestyle recommendations were provided to all participants randomized to receive metformin or placebo. The ILI consisted of an individualized curriculum focusing on nutrition, exercise and behavioral modification. A weight reduction of at least 7 percent of initial body weight was the goal for participants assigned to ILI. The DPP was conducted from 1996–2001 at 27 sites in the U.S. Racial ethnic minorities and women with prior GDM were prespecified target groups in the DPP recruitment protocol [20]. The study design, rationale, and outcomes have been described previously [21]. The institutional review board of the University of Illinois at Chicago reviewed the study protocol and determined this study exempt from human subjects research oversight.

### Study population

Diabetes Prevention Program participants were at least 25 years old, had a BMI $\geq$ 24 kg/m$^2$ ($\geq$ 22 kg/m$^2$ if Asian) and had prediabetes. Prediabetes was defined by a fasting glucose 95–125 mg/dL and a 2-hour 75-gram OGTT glucose 140–199 mg/dL. Participants were randomized to treatment with placebo, ILI, or metformin. Of note, women who were pregnant, less than 3 months postpartum, or currently nursing or within 6 weeks of completing nursing were excluded from the DPP trial. Women who answered the question, "Have you ever been told that you had a high sugar level or that you have diabetes?" and selected the answer "Only during pregnancy" were considered to have had GDM. The DPP study initially included a fourth intervention, troglitazone, which was discontinued in 1998 because of the drug's potential liver toxicity. Women with a history of GDM randomized to troglitazone were excluded from our analysis. Our study cohort included a subset of 317 women with prior GDM and prediabetes (Fig 1).

### Candidate predictor variables

Candidate predictor variables for the model included 11 baseline clinical variables known to be associated with diabetes progression: age group, ethnicity, parental (either mother or father) history of diabetes (type not specified), BMI group, waist circumference, waist-to-hip ratio, fasting glucose and triglycerides, HbA1c, self-reported physical activity, and treatment arm (placebo, ILI, or metformin). For variables with multiple readings at baseline (waist circumference and hip girth), an average was used. BMI was grouped as < 30, 30 to < 35, and $\geq$ 35 kg/m$^2$. The age of participants was available in 5-year intervals and collapsed at both extremes: age <40 and age $\geq$ 65 years. A family history of diabetes was determined if either parent had

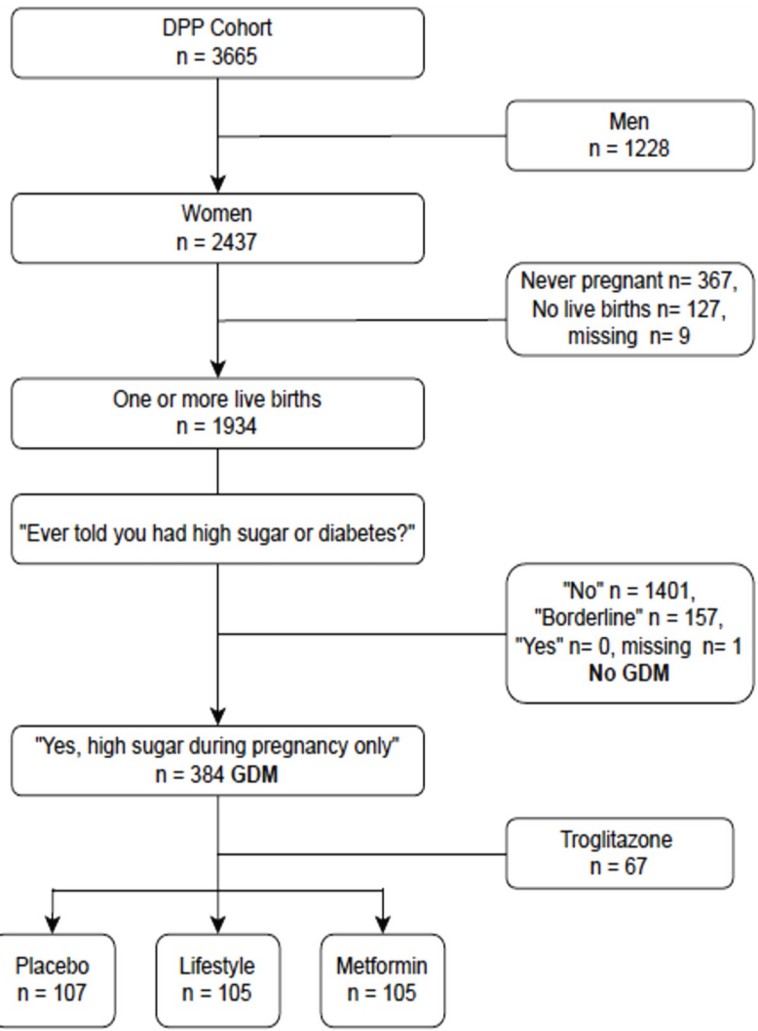

**Fig 1. Flow diagram for selection of women with GDM in the DPP cohort.**

diabetes. An unknown or negative parental history of diabetes was considered negative. Baseline leisure physical activity was assessed with a comprehensive list of activities in the DPP Modifiable Activity Questionnaire (MAQ), which estimated self-reported past-year activity by duration, frequency and relative intensity or metabolic equivalence task (MET) as expressed by MET-hours per week [22]. Physical activity estimates obtained from the MAQ correlated with measures of obesity and glucose tolerance in the DPP [23]. The 2011 Compendium of Physical Activity was used to determine the MET for each activity reported [24].

## Outcome measures

The outcome measure was the development of diabetes as defined by the ADA fasting glucose diagnostic cut off value, which in June 1997 was lowered to 126 mg/dL from 140 mg/dL [25]. Diagnostic criteria for diabetes was defined by a fasting plasma glucose $\geq$ 140 mg/dL (until June 23, 1997) or $\geq$ 126 mg/dL (on or after June 24, 1997), or a 2-hour 75-gram oral glucose tolerance test (OGTT) $\geq$ 200 mg/dL [21]. The diagnosis of diabetes was confirmed if consecutive testing with the same criteria, usually within 6 weeks, was met. The data was explored for

potential misclassification of diabetes due to the change in diagnostic criteria and we found the ultimate outcome of diabetes at three years was unaffected. All seventeen women who had two or more consecutive glucose readings between 126–140 mg/dL before June 23, 1997 were ultimately diagnosed with diabetes during the three-year follow-up period. Of the women who did not develop diabetes, none had two consecutive fasting glucoses $\geq$ 126 mg/dL. Clinical assessments were performed routinely every six months during the average three years of monitoring.

### Statistical analyses

**Bivariate analyses.** Descriptive statistics for continuous variables were expressed as median with interquartile range (IQR) due to our small sample size. Categorical variables were described as frequency and proportion. All statistical tests were two-sided. Fisher's exact and Kruskal Wallis were used, as appropriate for categorical and continuous variables, respectively, to test differences.

**Prediction risk model development and evaluation.** All models were developed by using multivariable Cox proportional hazards regression and informed by clinical knowledge and previous research [26, 27]. Continuous variables were standardized. Given the extreme right skewness of the fasting triglyceride data, the values were log-transformed to prevent large values from influencing the results. We determined, a priori, that a single model instead of a separate model for each treatment arm would be developed to maximize use of the subsampled DPP data. Residual diagnostic was performed for all models. Proportional hazard assumption for all covariates in all models were checked and satisfied. The 11 candidate variables, including treatment arm (placebo, ILI, or metformin), as well as interactions between treatment arm and BMI group, were entered into an initial model (Model 1). The interaction of treatment arm with BMI group was included because of observed heterogeneity in treatment effects with different BMI groups in the DPP study [8, 28]. We explored reduced models which included known, highly sensitive screening clinical variables (i.e. fasting glucose and HbA1c) and additional commonly accepted predictors (i.e. BMI group, fasting triglycerides) [10, 11, 29–31]. Finally, a parsimonious model (Model 2) was developed that only included significant predictors at 0.05 significance level from the bivariate analysis. Model 2 included screening clinical variables (HbA1c, fasting glucose), BMI group, treatment arm, and the BMI group by treatment arm interaction. To measure the discriminatory performance of the final parsimonious risk model, Harrell's C-statistic was computed with bias correction [32]. Internal validation was performed with 10-fold cross validation by randomly separating the data into 10 subsets, and estimating the parameters after omitting one of the 10 subsets. The model was applied to the omitted subset and Harrell's C statistics were calculated for discrimination for each omitted subset. All statistical analyses were performed using R version 3.6.1 (Survival, rms, Survminer, Caret, and survivalROC packages) and can be accessed through https://github.com/bsgerber/dpp-dppos.

## Results

### Bivariate analyses

Among 3665 adults with prediabetes, 317 (8.6%) women reported a history of GDM with treatment of placebo, ILI, or metformin. Table 1 shows the baseline characteristics of women with prior GDM by treatment arm. Among women with prior GDM: 61.5% were of reproductive age (< 45 years old) and 40.7% self-identified as non-Caucasian. Baseline characteristics were similar among the three treatments with the exception of waist-to-hip ratio (p = 0.02). Three years after randomization, 82 (25.9%) women developed diabetes (Table 2). Diabetes free survival curves for each treatment are provided in S1 Fig. The estimated incidence of diabetes

**Table 1. Baseline characteristics by treatment assignment.**

| | Lifestyle (N = 105) | Metformin (N = 105) | Placebo (N = 107) | p |
|---|---|---|---|---|
| **Age Group,** N (%) | | | | 0.13 |
| <40 | 38 (36.2%) | 38 (36.2%) | 39 (36.4%) | |
| 40–44 | 29 (27.6%) | 28 (26.7%) | 23 (21.5%) | |
| 45–49 | 23 (21.9%) | 21 (20.0%) | 30 (28.0%) | |
| 50–54 | 5 (4.8%) | 11 (10.5%) | 14 (13.1%) | |
| 55–59 | 5 (4.8%) | 2 (1.9%) | 1 (0.9%) | |
| 60+ | 5 (4.8%) | 5 (4.8%) | 0 (0.0%) | |
| **Ethnicity,** N (%) | | | | 0.77 |
| Caucasian | 64 (61.0%) | 66 (62.9%) | 58 (54.2%) | |
| African American | 18 (17.1%) | 19 (18.1%) | 26 (24.3%) | |
| Hispanic, of any race | 18 (17.1%) | 16 (15.2%) | 20 (18.7%) | |
| All other | 5 (4.8%) | 4 (3.8%) | 3 (2.8%) | |
| **Smoking Status,** N (%) | | | | 0.28 |
| Current | 8 (7.6%) | 4 (3.8%) | 5 (4.7%) | |
| Former | 27 (25.7%) | 22 (21.0%) | 34 (31.8%) | |
| ≤ 100 cig lifetime | 70 (66.7%) | 79 (75.2%) | 68 (63.6%) | |
| **PCOS History,** N (%) | | | | 0.37 |
| Yes | 2 (1.9%) | 0 (0.0%) | 2 (1.9%) | |
| No | 103 (98.1%) | 105 (100.0%) | 105 (98.1%) | |
| **BMI Group kg/m$^2$,** N (%) | | | | 0.55 |
| < 30 | 29 (27.6%) | 30 (28.6%) | 25 (23.4%) | |
| 30 to < 35 | 33 (31.4%) | 41 (39.0%) | 37 (34.6%) | |
| 35+ | 43 (41.0%) | 34 (32.4%) | 45 (42.1%) | |
| **Family History of Diabetes,** N (%) | 57 (54.3%) | 53 (50.5%) | 71 (66.4%) | 0.05 |
| **Live Births,** Median (IQR) | 2.0 (2.0, 3.0) | 2.0 (2.0, 3.0) | 2.0 (2.0, 3.0) | 0.99 |
| **Waist Circumference (cm),** Median (IQR) | 102.8 (93.0, 111.5) | 98.3 (91.8, 107.3) | 102.5 (91.1, 108.9) | 0.18 |
| **Waist to Hip Ratio,** Median (SD) | 0.881 ((0.842, 0.929) | 0.867 ((0.823, 0.912) | 0.872 (0.836, 0.920) | 0.07 |
| **Fasting Glucose (mg/dL),** Median (IQR) | 105 (100, 110) | 105 (100, 112) | 106 (101, 112) | 0.49 |
| **Hemoglobin A1c (%),** Median (IQR) | 5.9 (5.5, 6.1) | 5.8 (5.6, 6.1) | 5.9 (5.5, 6.2) | 0.66 |
| **MET-hours/week,** Median (IQR) | 10.6 (4.2, 18.6) | 8.9 (4.0, 20.0) | 8.5 (4.1, 16.2) | 0.58 |
| **Systolic BP (mmHg),** Median (IQR) | 118 (109, 126) | 115 (108, 124) | 117 (109, 126) | 0.60 |
| **Diastolic BP (mmHg),** Median (IQR) | 75 (70, 80) | 76 (70, 81) | 75 (70, 82) | 0.98 |
| **Triglyceride (mg/dL),** Median (IQR) | 136 (97, 204) | 119 (94, 179) | 130 (98, 193) | 0.46 |

Column percentages are presented.

*Abbreviations*: Cig cigarettes; PCOS Polycystic Ovarian Syndrome; BMI Body Mass Index; MET Metabolic Equivalent Task; IQR Interquartile Range; BP Blood Pressure.

without treatment (placebo group) was 37.4% for women with prior GDM, compared with 20% for either ILI or metformin (p < 0.01).

## Risk models

Cox proportional hazard models for Models 1 and 2 are presented in Table 3. Using model 1, fasting glucose and HbA1C were always significant predictors of higher hazard. Women with

**Table 2. Baseline characteristics by diabetes outcome.**

| | No Diabetes (N = 235) | Diabetes (N = 82) | p |
|---|---|---|---|
| **Age Group,** N (%) | | | 0.44 |
| <40 | 88 (37.4%) | 27 (32.9%) | |
| 40–44 | 59 (25.1%) | 21 (25.6%) | |
| 45–49 | 55 (23.4%) | 19 (23.2%) | |
| 50–54 | 18 (7.7%) | 12 (14.6%) | |
| 55–59 | 6 (2.6%) | 2 (2.4%) | |
| 60+ | 9 (3.8%) | 1 (1.2%) | |
| **Ethnicity,** N (%) | | | 0.14 |
| Caucasian | 148 (63.0%) | 40 (48.8%) | |
| African American | 44 (18.7%) | 19 (23.2%) | |
| Hispanic, of any race | 35 (14.9%) | 19 (23.2%) | |
| All other | 8 (3.4%) | 4 (4.9%) | |
| **Smoking Status,** N (%) | | | 0.97 |
| Current | 13 (5.5%) | 4 (4.9%) | |
| Former | 61 (26.0%) | 22 (26.8%) | |
| ≤100 cig lifetime | 161 (68.5%) | 56 (68.3%) | |
| **PCOS History,** N (%) | | | 0.23 |
| Yes | 4 (1.7%) | 0 (0.0%) | |
| No | 231 (98.3%) | 82 (100.0%) | |
| **BMI Group (kg/m$^2$),** N (%) | | | 0.19 |
| <30 | 59 (25.1%) | 25 (30.5%) | |
| 30 to <35 | 89 (37.9%) | 22 (26.8%) | |
| 35+ | 87 (37.0%) | 35 (42.7%) | |
| **Family History of Diabetes,** N (%) | 133 (56.6%) | 48 (58.5%) | 0.76 |
| **Live Births,** Median (IQR) | 2.0 (2.0, 3.0) | 2.0 (2.0, 3.0) | 0.88 |
| **Waist Circumference (cm),** Median (IQR) | 100.0 (92.0, 107.8) | 103.9 (91.9, 112.6) | 0.47 |
| **Waist to Hip Ratio,** Median (SD) | 0.876 (0.835, 0.914) | 0.884 (0.837, 0.935) | 0.43 |
| **Fasting Glucose (mg/dL),** Median (IQR) | 104 (100, 110) | 111 (106, 118) | < 0.01 |
| **Hemoglobin A1c (%),** Median (IQR) | 5.8 (5.5, 6.1) | 6.1 (5.7, 6.3) | < 0.01 |
| **MET-hours/week,** Median (IQR) | 9.0 (4.0, 18.5) | 9.4 (4.8, 19.8) | 0.58 |
| **Systolic BP (mmHg),** Median (IQR) | 116 (108, 126) | 118 (111, 127) | 0.20 |
| **Diastolic BP (mmHg),** Median (IQR) | 76 (70, 80) | 74 (70, 83) | 0.58 |
| **Triglyceride (mg/dL),** Median (IQR) | 129 (98, 186) | 112 (91, 209) | 0.70 |
| **Treatment Arm** | | | <0.01 |
| Placebo | 67 (28.5%) | 40 (48.8%) | |
| Lifestyle | 84 (35.7%) | 21 (25.6%) | |
| Metformin | 84 (35.7%) | 21 (25.6%) | |

Column percentages are presented, row percentages are available in S1 Table.

*Abbreviations*: Cig cigarettes; PCOS Polycystic Ovarian Syndrome; BMI Body Mass Index; MET Metabolic Equivalent Task; IQR Interquartile Range; BP Blood Pressure.

a BMI $\geq$ 35 kg/m$^2$ had the greatest risk of progressing to diabetes in the placebo arm (S2 Fig). Both models were evaluated and found to have fair discriminative performance corrected with ten-fold cross validation (Table 3). Using likelihood-ratio tests, we assessed the goodness of fit of Models 1 and 2 (rms package) and found no significant differences in discriminative performance. We include the diabetes prediction equation based on Model 2 in Table 4.

**Table 3. Two models predicting progression to diabetes at three years for women with prior gestational diabetes.**

| Baseline Predictors[†] | Hazard Ratio (95% CI) | |
|---|---|---|
| | Model 1 | Model 2 |
| **Treatment effect in normal BMI group** | | |
| Placebo | | |
| Lifestyle | 0.404 (0.121, 1.346) | 0.527 (0.163, 1.709) |
| Metformin | 1.108 (0.447, 2.751) | 1.210 (0.510, 2.869) |
| **Fasting glucose, mg/dL** | 1.537* (1.233, 1.914) | 1.465* (1.198, 1.793) |
| **Hemoglobin A1c, %** | 1.449* (1.096, 1.917) | 1.379* (1.083, 1.755) |
| **BMI group, kg/m$^2$, effect in placebo group** | | |
| < 30 | | |
| ≥ 30 to < 35 | 0.678 (0.251, 1.829) | 1.049 (0.442, 2.486) |
| ≥ 35 | 0.537 (0.180, 1.605) | 1.212 (0.564, 2.607) |
| **Log triglycerides, mg/dL** | 0.953 (0.740, 1.228) | |
| **Race/Ethnicity** | | |
| Caucasian | | |
| African American | 0.746 (0.372, 1.498) | |
| Hispanic | 1.723 (0.934, 3.179) | |
| Other | 1.647 (0.539, 5.031) | |
| **Age group, years** | | |
| < 40 | | |
| 40–44 | 1.022 (0.521, 2.005) | |
| 45–49 | 1.370 (0.733, 2.560) | |
| 50–54 | 1.483 (0.702, 3.133) | |
| 55–59 | 1.144 (0.220, 5.938) | |
| 60+ | 0.731 (0.093, 5.739) | |
| **Family History** | 0.918 (0.575, 1.466) | |
| **Waist circumference, cm** | 1.426 (0.955, 2.130) | |
| **Waist-to-hip ratio** | 1.066 (0.820, 1.385) | |
| **MET-hours/week** | 1.166 (0.931, 1.460 | |
| **BMI ≥ 30 to < 35 kg/m$^2$** | | |
| Lifestyle vs Placebo | 1.627 (0.345, 7.676) | 1.246 (0.274, 5.667) |
| Metformin vs Placebo | 0.291 (0.066, 1.289) | 0.257 (0.061, 1.083) |
| **BMI ≥ 35 kg/m$^2$** | | |
| Lifestyle vs Placebo | 1.175 (0.286, 4.830) | 1.017 (0.251, 4.115) |
| Metformin vs Placebo | 0.281 (0.072, 1.095) | 0.247* (0.071, 0.858) |
| R$^2$ | 0.191 | 0.151 |
| Bias corrected C | 0.6577 | 0.6868 |

[†]continuous variables are standardized.

*$p < 0.05$

*Abbreviations*: CI confidence intervals; BMI body mass index.

Additionally, time dependent receiver operating characteristic (ROC) curves were produced for both models (S3 Fig).

## Discussion

A diabetes risk prediction model using four commonly-available clinical measures was successfully developed for women with prior GDM. Our model performed with similar

**Table 4. Model equation[†] to calculate probability of developing diabetes at 3 years.**

| Probability of progression to DM = $1-S0(t)^{exp(f(x))}$ |
| --- |
| S0(3 years) = 0.656 |
| F(X) = |
| - 0.640 x TREATMENT$_L$ |
| + 0.191 x TREATMENT$_M$ |
| + 0.047 x BMI$_1$ |
| + 0.193 x BMI$_2$ |
| + 0.382 x ((FASTING GLUCOSE– 107.1293) / 7.4786) |
| + 0.321 x ((HEMOGLOBIN A1c- 5.8427) /0.4834) |
| + 0.220 x TREATMENT$_L$ x BMI$_1$ |
| + 0.017 x TREATMENT$_L$ x BMI$_2$ |
| - 1.358 x TREATMENT$_M$ x BMI$_1$ |
| - 1.400 x TREATMENT$_M$ x BMI$_2$ |

[†]S0 = 3-year survival probability for a woman with the reference covariate pattern where the categorical covariates are set at their reference (placebo, BMI group < 30) and continuous variables are set at the mean (continuous variables are standardized).

TREATMENT$_L$ 1 if treatment is lifestyle, 0 otherwise
TREATMENT$_M$ 1 if treatment is metformin, 0 otherwise
BMI$_1$ 1 if BMI $\geq$ 30 to <35 kg/m$^2$, 0 otherwise
BMI$_2$ 1 if BMI $\geq$ 35 kg/m$^2$, 0 otherwise

discrimination as other diabetes prediction models developed from the DPP for the general population [10, 11]. Similar to other models derived from the general population, our parsimonious model includes fasting glucose, HbA1c, BMI, and treatment (ILI, metformin, or placebo) [33]. These commonly available measures offer great value to clinicians in evaluating diabetes risk. Measurement of HbA1c is a guideline-recommended, routinely-assessed screening test for diabetes, and is known to predict diabetes in the DPP cohort and in other large community-based cohorts [25, 34, 35]. In addition, a woman's BMI is a modifiable, strongly-predictive risk factor for diabetes after a GDM pregnancy [19]. Although measures of abdominal obesity, waist circumference and waist-to-hip ratio correlate with cardiometabolic risk, neither added predictive value in our models [33, 36]. Furthermore, the addition of the triglyceride level, a clinical marker of metabolic syndrome and insulin resistance, did not improve discrimination [29, 30, 37, 38].

Other models have identified similar risk factors for diabetes among women with GDM [18, 19, 39]. In a prospective case-control study of 150 Australian women with GDM and 72 overweight women with normal glucose tolerance, GDM women with a high risk profile (elevated BMI, blood pressure, glucose, insulin, triglyceride, and lower high-density lipoprotein levels) were more likely to develop diabetes compared to women with a low-risk profile in a cluster analysis [40]. Among 1,263 Chinese women with prior GDM, waist circumference, body fat and BMI were all associated with an increased risk of diabetes, with waist circumference and body fat better indicators for diabetes than BMI [41]. Women with previous GDM have a high-risk profile similar to metabolic syndrome where there may be greater benefit from risk reduction therapy [42].

Although a number of risk factors for diabetes are well known, risk estimation is not commonly used in clinical practice. Risk prediction models can facilitate medical decision making [43], and may be more accurate or, at least, less biased than subjective predictions. An

evidence-based GDM risk estimate can provide individualized health information to women and clinicians to guide prediabetes treatment discussion. Model equations (Table 4) may help clinicians estimate the risk of progression to diabetes and magnitude of risk reduction with different prediabetes treatment options (i.e., metformin and/or ILI). To realize the benefits of risk prediction, clinical implementation of decision-making activities including aids must be further investigated. Barriers to utilizing risk calculators have been described and may include concern for generalizability and lack of added value to clinical judgment [44, 45].

It is estimated that women with prior GDM have at least a seven-fold increased risk of developing type 2 diabetes compared with those with normoglycemic pregnancies.[46] The model's modifiable clinical measures appear most relevant to consider in diabetes risk assessment of women with prior GDM, particularly when incorporated into treatment decision making activities. We have developed a decision aid targeting women with prior GDM to assist in decisions regarding diabetes prevention therapy. Models such as those derived from the DPP may be incorporated into decision aids to provide users with calculated diabetes risk scores, if treated with ILI, metformin, or standard lifestyle recommendations.

We recognize our study has a number of limitations. First, this model was developed from DPP participants, and may not be representative of the general population of women with GDM. Adults with certain chronic medical diseases and those taking common chronic medications were excluded from the DPP trial. The diagnosis of GDM was based on self-reporting; however, studies show women recall their GDM diagnosis and treatment accurately [47, 48]. Pregnant women with undiagnosed, preexisting diabetes may be inadvertently diagnosed with GDM, if early first trimester screening is not performed [49]. However, women with prepregnancy diabetes will likely have diabetes after delivery and would have been excluded from the DPP. In addition, women with prior GDM with subsequent prediabetes may not have selected the answer "only during pregnancy" when asked about a history of hyperglycemia; these women were not in our sample.

Women with GDM are at greatest risk of developing diabetes within the postpartum period and first five years of their index pregnancy; the mean time from the index pregnancy was 12 years for GDM women in the DPP [50]. Women who were in the immediate postpartum period and/or breastfeeding were excluded and the mean time from the index pregnancy was 12 years for GDM women in the DPP [50]. Thus, women enrolled in the DPP may not have represented women with the greatest risk for diabetes, and the model may underestimate the predicted risk of diabetes in women with GDM. Second, prediabetes and incident diabetes were defined by fasting blood glucose and 75-gram OGTT criteria in the DPP. In clinical practice, however, screening and diagnosis are more commonly performed with HbA1c. Interestingly, risk reduction by metformin and lifestyle were similar, 44% and 49% respectively, when diabetes incidence was defined by HbA1c $\geq$ 6.5% (48 mmol/mol) for the overall (men and women) DPP cohort [35]. Third, estimates of risk reduction with an ILI are based on the specific DPP intensive lifestyle program. The risk reduction benefit may not apply to other less-intensive lifestyle interventions. Fourth, our model does not account for treatment adherence (metformin or lifestyle), which would contribute to heterogeneity in treatment effects. Anticipated adherence to therapy is important in decision making, as lifestyle adherence is more effective than metformin in promoting regression to normal glucose regulation [11]. Lastly, our model does not account for peripartum related predictors (e.g. pregnancy OGTT levels or postpartum BMI) which may be more predictive of diabetes risk, though these measures are not always readily available at future medical encounters. Finally, the model was developed utilizing only GDM participants in the DPP. Given the relatively small sample meeting our criteria, external validation is necessary before clinical use.

## Conclusion

We have developed and internally validated a clinically applicable prediction model which includes fasting glucose, HbA1c, BMI, treatment arm, and BMI by treatment arm interaction for women with prior GDM. Incorporating individualized diabetes risk prediction into prediabetes treatment decision making may improve understanding the potential benefit of ILI and/or metformin in diabetes prevention.

## Supporting information

**S1 Fig. Kaplan Meier curves for probability of diabetes free with follow up time of 3 years of women with prediabetes and a history of gestational diabetes in placebo, intensive lifestyle, and metformin arms.**
(TIF)

**S2 Fig. Predicted probability of not progressing to diabetes by BMI group and treatment.**
(TIF)

**S3 Fig. Time dependent receiver operating characteristic (ROC) curves for model 1 (full) and model 2 (parsimonious).**
(TIF)

**S1 Table. Baseline characteristics by diabetes outcomes with row percent.**
(DOCX)

## Acknowledgments

The Diabetes Prevention Program (DPP) was conducted by the DPP Research Group and supported by the National Institute of Diabetes and Digestive and Kidney Diseases (NIDDK), the General Clinical Research Center Program, the National Institute of Child Health and Human Development (NICHD), the National Institute on Aging (NIA), the Office of Research on Women's Health, the Office of Research on Minority Health, the Centers for Disease Control and Prevention (CDC), and the American Diabetes Association. The Diabetes Prevention Program dataset supporting the conclusions of this article is available upon request in the National Institute of Diabetes and Digestive and Kidney Diseases Central Repositories, https://repository.niddk.nih.gov/studies/dpp/. This manuscript was not prepared under the auspices of the DPP and does not represent analyses or conclusions of the DPP Research Group, the NIDDK Central Repositories, or the National Institute of Health.

## Author Contributions

**Conceptualization:** Bernice Man, Alan Schwartz, Ben Gerber.

**Formal analysis:** Bernice Man, Alan Schwartz, Oksana Pugach, Yinglin Xia, Ben Gerber.

**Investigation:** Bernice Man, Ben Gerber.

**Methodology:** Bernice Man, Alan Schwartz, Yinglin Xia, Ben Gerber.

**Supervision:** Alan Schwartz, Oksana Pugach, Ben Gerber.

**Writing – original draft:** Bernice Man, Ben Gerber.

**Writing – review & editing:** Bernice Man, Alan Schwartz, Oksana Pugach, Yinglin Xia, Ben Gerber.

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
