## [Decision Letter · Decision Letter 0]

30 Dec 2020

PONE-D-20-34391

A Clinical Diabetes Risk Prediction Model For Prediabetic Women With Prior Gestational Diabetes

PLOS ONE

Dear Dr. Man,

Thank you for submitting your manuscript to PLOS ONE. After careful consideration, we feel that it has merit but does not fully meet PLOS ONE’s publication criteria as it currently stands. Therefore, we invite you to submit a revised version of the manuscript that addresses the points raised during the review process.

We look forward to receiving your revised manuscript.

Kind regards,

Andreas Beyerlein

Academic Editor

PLOS ONE

Additional Editor Comments:

- I appreciate it very much that the authors aim to make their data fully available. However, the link of the data repository seems to require a personal registration beforehand. Personally, I think this is OK, but it should not be labelled as "data are fully available without restriction".

- Furthermore, the authors should also make their data analysis code available.

- Abstract: It does not get clear that this was a secondary analysis of a clinical trial. For example, the fact that 82 women developed diabetes within three years and the estimated incidences in the groups with and without treatment are not a result of this particular study and should therefore be mentioned in the Methods. It should also be mentioned there how treatment and control groups were defined. Further, the results should be more specific with respect to the associations between the individual predictors and the diabetes risk.

- Was there no information about GDM treatment and breastfeeding available? Both were relevant predictors of postpartum diabetes in an analysis of Köhler et al., Acta Diabetol 2016.

- Further I wonder why the mean time between the first (or last) pregnancy with GDM and study enrolment and the number of diabetic pregnancies were not investigated as potential predictors? At least, both variables should be mentioned in table 1.

- How was family history of diabetes defined? Could it be differed between GDM, type 1 and type 2 diabetes?

- l. 168-170: I suggest to use median and IQR throughout and not to mix it with mean / SD.

- l. 171-173: Fisher's exact test should be used instead of Chi-Square tests when expected cell numbers are too low.

- l. 177-178: Please give another rationale for log-transformation of fasting triglycerides as normal distribution is not a requirement for predictor variables.

- l. 181-191: This approach makes no sense to me. A more common approach would be to start with either all potentially relevant variables (i.e. model 1) or only those variables which were significant in the bivariate analyses (i.e. start with model 3) and then (in either case) reduce the number of variables through variable selection.

- l. 195: Where exactly was cross validation applied and for which purpose?

- Figure 1: Please explain the meaning of the arrow "Troglitazone"

- Tables 1 and 2: It should be mentioned that the % values are column %. However, I think row % might be more appropriate in table 2.

- Table 3: I doubt that the interaction terms of BMI and intervention groups are of use here because the 95% confidence intervals of the respective hazard ratios are so wide. Furthermore, the hazard ratios of the basic variables treatment arm and BMI cannot be intepreted for themselves when the models include the respective interaction terms.

- l. 217-218: How was significance determined in this sentence?

- Table 4 needs more explanation, e.g. what is the meaning of S0, or where do the factors in the fasting glucose and HbA1c lines come from?

- Suggest to show a Kaplan-Meier plot in Suppl. Figure 1.

- Please dicuss how the risk score compares to those from other studies in terms of predicitive performance.

Journal Requirements:

Reviewers' comments:

Reviewer's Responses to Questions

**Comments to the Author**

1. Is the manuscript technically sound, and do the data support the conclusions?

Reviewer #1: Yes

Reviewer #2: Yes

2. Has the statistical analysis been performed appropriately and rigorously? 

Reviewer #1: Yes

Reviewer #2: Yes

3. Have the authors made all data underlying the findings in their manuscript fully available?

Reviewer #1: Yes

Reviewer #2: Yes

4. Is the manuscript presented in an intelligible fashion and written in standard English?

Reviewer #1: Yes

Reviewer #2: Yes

5. Review Comments to the Author

Reviewer #1: A secondary analysis of the DPP trial which included 317 women with gestational diabetes.

It is an intriuging study but with some caveats: First the risk model is practical and simple and may be used on a primary conselling setting. the simplicity is maybe also the piont of where exact values of risk are not to be trusted: It was a sample of study not geared for pregnancy and the dioagnosis of GDM is based on self-reported statements on what was told during pregnancy. Knowing that glucosuria is a copmmon recurrence in normal pregnancy , too, some misclassification could be present. For second the relative few women to make a model on 5 variables seem to stretch the creditability. Unless, of course, the correlations are that strong that even in each substrata varibales exerts its effect on the outcome. A hint that the sample of 317 may be too skewed is the non-significance of ethnicity. I think too many caucasians with certain traits have been included in the DPP and, consequently, fewer of the ethnic mixed with other diabetogenic backgrounds. Nevertheless, it may be true and work under these circumstances!

Smaller issues:

They introduce the subject by claiming equipotency of metformin and ILI in that both have 50 % reduction diabetes risk; well may be, if applied on similar time shortly before diabetes would manifest itself: while LIL may reduce the risk anytime, metformin is most potent right before the beta-cell goes dry and diabetes would manifest itself

titel page: the affilations denoted as superscripts on the authors names (1-5) do not correspond to the institutions (a-d)

Reviewer #2: This paper presents important information that adds to the GDM knowledge pool, especially in the risk prediction model of subsequent prediabetes or diabetes. The paper is well written, with a clear text, easy to read. However, I have few comments sated as follows:

1. Method section of the abstract: Though, developed and validated a 3-year diabetes prediction model using Cox proportional hazards regression acceptable but the risk prediction model using 11 baseline clinical variables independently and/ or in combined for risk of prediabetes would be computed by the area under the ROC curve (discriminative power). In addition, a risk score calculation based on a combination of the most pertinent clinical variables model prediction for prediabetes is highly suggested.

2. As insulin also given for GDM, why the prediction model is not included about insulin management or why authors focused only lifestyle intervention vs metformin treatment?

3. There is controversies in some statement in the use of prediabetes vs diabetes eg. the title A Clinical Diabetes Risk Prediction Model For Prediabetic Women With Prior Gestational Diabetes. Meaning that the main outcome focused on “prediabetes”, However the result section of the abstract “ Within three years, 82 (25.9%) women developed diabetes” and Conclusion section of the abstract also “clinical prediction model was developed for individualized decision making for prediabetes treatment in women with prior GDM” . As Prediabtes and Diabetes has different definition and cutoff point of blood glucose levels, the use of the two terms should be managed thorough out the entire document.

4. The introduction section page 6 line 91-105: What is the relevance of the stated sentences about model comparison here? I suggested to remove this and described in detail in the method section.

5. Page 7 line 111 “ ------ neither (placebo)” and result section page 13 Table 1 “Placebo (N=107)” How could the GDM patient in placebo. They should get at least the minimum standard treatment. It has also the ethical concerns?? Clarification is needed

6. Method section Page 7 line 120 “ --- placebo on the development of diabetes over an average of 3.2 years” Confusing ???

7. Page 8 line 134-136:“Women who answered the question, “Have you ever been told that you had a high sugar level or that you have diabetes” and selected the answer “Only during pregnancy” were considered to have had GDM”. What if there is undiagnosed preexisting diabetes mellitus?

8. As authors stated “ The outcome measure was the development of diabetes as defined by ADA criteria.( Clinical assessments were performed routinely every six months during the average three years of monitoring. Diagnostic criteria for diabetes was defined by a fasting plasma glucose ≥ 140 mg/dL (until June 23, 1997) or ≥ 126 mg/dL (on or after June 24, 1997), or a 2-hour 75-gram oral glucose tolerance test (OGTT) ≥ 200 mg/dL.(20) The diagnosis of diabetes was confirmed if consecutive testing with the same criteria, usually within 6 weeks, was met.” The Outcome Measures ascertainment is confusing eg. FPG ≥ 140 mg/dL or ≥ 126 mg/dL ???? This leads missclassifications bias. Clarification is needed

9. Method section: ROC analysis for “Prediction risk model development and evaluation” and its main findings eg. AUC in the result section and supported by figures also suggested.

10. Adding the implication of main findings has recommend to make stronger discussion section

11. Discussion section. Page 21 line 252-264. The paragraphs is not related with your findings eg. Plasma lipidomic analysis, common genetic loci…

6. PLOS authors have the option to publish the peer review history of their article (what does this mean?). If published, this will include your full peer review and any attached files.

Reviewer #1: **Yes: **Finn Lauszus

Reviewer #2: **Yes: **Achenef Asmamaw Muche

---

## [Author Response · Author response to Decision Letter 0]

13 Apr 2021

We have provided this as a separate attachment but included it in its entirety here. 

Dear Plos One Reviewers,

Thank you for giving us the opportunity to submit a revised draft of the manuscript “A Clinical Diabetes Risk Prediction Model For Prediabetic Women With Prior Gestational Diabetes” for publication in Plos One. We appreciate the time and effort that the reviewers dedicated to providing feedback on our manuscript and are grateful for the insightful comments on and valuable improvements to our paper. We have incorporated most of the suggestions made by the reviewers. Those changes are highlighted within the manuscript. Please see below, in blue, for a point-by-point response to the reviewers’ comments and concerns. All page numbers refer to the revised manuscript file with tracked changes. 

1)- I appreciate it very much that the authors aim to make their data fully available. However, the link of the data repository seems to require a personal registration beforehand. Personally, I think this is OK, but it should not be labelled as "data are fully available without restriction".

We have modified the data availability statement to include that access to DPP data is available upon request to the National Institute of Diabetes and Digestive and Kidney Diseases Central Repositories (https://repository.niddk.nih.gov/studies/dpp/) in the manuscript (line 101), acknowledgements (line 479) and cover letter. 

2)- Furthermore, the authors should also make their data analysis code available.

Currently, our data analysis code is maintained in a private repository on Github. We will change this to a public repository upon acceptance of the manuscript.

3)- Abstract: It does not get clear that this was a secondary analysis of a clinical trial. For example, the fact that 82 women developed diabetes within three years and the estimated incidences in the groups with and without treatment are not a result of this particular study and should therefore be mentioned in the Methods. It should also be mentioned there how treatment and control groups were defined. Further, the results should be more specific with respect to the associations between the individual predictors and the diabetes risk.

For the methods in the abstract (lines 7-13), we have added that this is a secondary analysis of the DPP and defined the treatment and control groups. For the abstract results, we included the variables (both main and conditional effects) of our parsimonious model but have chosen to not include the hazard ratios and 95% confidence intervals for simplicity (line 15-19).

4)- Was there no information about treatment and breastfeeding available? Both were relevant predictors of postpartum diabetes in an analysis of Köhler et al., Acta Diabetol 2016. - Further I wonder why the mean time between the first (or last) pregnancy with GDM and study enrolment and the number of diabetic pregnancies were not investigated as potential predictors? At least, both variables should be mentioned in table 1.

Thank you for the additional reference. We have added the above article to our citations and note the predictors (lines 74, 77-78) in our introduction. There were no DPP survey questions about breast feeding history or GDM treatment for us to explore as potential predictors. While the survey asked about the total number of live births, which is included in Table 1, there was no question about the number of pregnancies with diabetes. In addition, women who were pregnant, less than 3 months postpartum, or currently nursing or within 6 weeks of completing nursing were excluded from the DPP trial. These exclusions were noted in our discussion of study limitations (lines 430-432) previously and we now also note the exclusions in our methods (lines 125-127) .

5)- How was family history of diabetes defined? Could it be differed between GDM, type 1 and type 2 diabetes?

Family history was defined as a binary indicator variable: (1) either mother or father with diabetes or (2) no known family history. We clarified this in the revision (lines 140-141, 147-148) . The type of diabetes a family member had was not specified in the survey. The survey question asked “Did your mother or father have diabetes?” 

6)- l. 168-170: I suggest to use median and IQR throughout and not to mix it with mean / SD.

In Table 1, we now use median and IQR to describe waist-to-hip ratio to be consistent with the other continuous variables. 

7)- l. 171-173: Fisher's exact test should be used instead of Chi-Square tests when expected cell numbers are too low.

A few variables did have expected cell counts <5. As suggested, we have revised the comparisons for all the categorial variables in Table 1 to use the Fisher’s exact test instead of Chi Square for conservative comparisons. 

8)- l. 177-178: Please give another rationale for log-transformation of fasting triglycerides as normal distribution is not a requirement for predictor variables.

We have added further clarification for the log transformation of triglyceride values which were extremely right skewed (lines 186-188).

9)- l. 181-191: This approach makes no sense to me. A more common approach would be to start with either all potentially relevant variables (i.e. model 1) or only those variables which were significant in the bivariate analyses (i.e. start with model 3) and then (in either case) reduce the number of variables through variable selection.

Model 1 was our initial and main model in which we started with available relevant variables informed by clinical knowledge and previous research (lines 184-186). The common approaches to variable selection (i.e., backward elimination or forward selection) are reliable but may suffer from systematic biases.(see references below) Our initial manuscript Models 2 and 3 were reduced models in which we further explored the role of well-established, known risk factors (e.g., race/ethnicity), and commonly available clinical factors (e.g, fasting triglycerides). We have removed model 2 from the manuscript given its development was not based on the traditional approaches for variable selection. However, we include our parsimonious model (revised Model 2), based on bivariate analysis, and its equation for diabetes risk calculation because of its potential clinically utility (lines 201- 211). 

References:

Sauerbrei et al. State of the art in selection of variables and functional forms in multivariable analysis—outstanding issues. Diagnostic and Prognostic Research (2020) 4:3 https://www.ncbi.nlm.nih.gov/pmc/articles/PMC7114804/

Heinze et al. Variable selection - A review and recommendations for the practicing statistician. Biom J 2018 May;60(3):431-449. https://www.ncbi.nlm.nih.gov/pmc/articles/PMC5969114/

10)- l. 195: Where exactly was cross validation applied and for which purpose?

Estimating discriminatory performance (Harrell’s C) without overfitting requires separating the data into model building and model testing subsets, which we did through 10-fold cross validation. Cross validation was applied for internal validation of the models. We randomly separated the data into 10 subsets, and estimated the parameters after omitting 1 of the 10 subsets. Then we applied the model to the omitted subset. Harrell’s C-statistics were calculated for discrimination for each omitted subset. This is clarified in the methods section (lines 213- 217).

11)- Figure 1: Please explain the meaning of the arrow "Troglitazone"

The DPP study initially included a fourth intervention, troglitazone, which was discontinued in 1998 because of the drug's potential liver toxicity. Women with a history of GDM randomized to troglitazone were excluded from our analysis. We have added this exclusion criteria to our description of Figure 1 with respect to subject selection in the methodology (lines 130- 133).

12)- Tables 1 and 2: It should be mentioned that the % values are column %. However, I think row % might be more appropriate in table 2.

We clarify that column percentages are included in Tables 1 and 2. For Table 2, we believe this is preferable as we are comparing the distribution of factors between two populations (based on outcome of whether developed diabetes or not). We have provided a Supplementary Table 1 to show the row percentages for Table 2 (noted in footnotes of Table 2).

Reference: https://www.annalsthoracicsurgery.org/article/S0003-4975(15)01520-9/fulltext

13)- Table 3: I doubt that the interaction terms of BMI and intervention groups are of use here because the 95% confidence intervals of the respective hazard ratios are so wide. Furthermore, the hazard ratios of the basic variables treatment arm and BMI cannot be interpreted for themselves when the models include the respective interaction terms.

The confidence intervals for the interaction terms are appropriate. The interactions for our exploratory models are significant which is why we felt it was better to keep them in the prediction model (Model 2). Although the regression coefficients of the terms that are part of the interaction cannot be interpreted as main effects, instead they should be interpreted as conditional effects. The effect of BMI will vary depending on the treatment group.

14)- l. 217-218: How was significance determined in this sentence?

We clarified the use of likelihood-ratio tests to compare models in discriminative performance (lines 280-282).

15)- Table 4 needs more explanation, e.g. what is the meaning of S0, or where do the factors in the fasting glucose and HbA1c lines come from?

S0 is a baseline survivor function, in our case, the probability of diabetes by 3 years, where baseline is defined as a set of categorical covariates at their reference categories and at zero for continuous covariates. Since all continuous variables were standardized, zero represents the mean of a variable. The factors in the fasting glucose and HbA1c are the sample means and standard deviations of the respective variables. We have added this to the footnote for Table 4.

16)- Suggest show a Kaplan-Meier plot in Suppl. Figure 1.

We have added Supplementary Figure 1, a Kaplan Meier curve, to show the probability of diabetes free with a follow-up of three years. 

17)- Please discuss how the risk score compares to those from other studies in terms of predictive performance.

We added to the discussion clarification of how our prediction model compares to those from other studies in terms of predictive performance (lines 337-339). Our model performed with similar discrimination as other diabetes prediction models developed from the DPP for the general population. 

Journal Requirements:

Check file naming

The DPP data is publicly available upon request to the National Institute of Diabetes and Digestive and Kidney Diseases Central Repositories, https://repository.niddk.nih.gov/studies/dpp/.

We have amended acknowledgements to state: The DPP data is publicly available upon request to the National Institute of Diabetes and Digestive and Kidney Diseases Central Repositories, https://repository.niddk.nih.gov/studies/dpp/.

 The author affiliations have been amended, updated and linked to the appropriate authors’ names.

Captions for supplementary figures 1, 2, and 3 and table 1 were added to the end of the manuscript.

Comments to the Author

1. Is the manuscript technically sound, and do the data support the conclusions?

Reviewer #1: Yes

Reviewer #2: Yes

2. Has the statistical analysis been performed appropriately and rigorously?

Reviewer #1: Yes

Reviewer #2: Yes

3. Have the authors made all data underlying the findings in their manuscript fully available?

Reviewer #1: Yes

Reviewer #2: Yes

4. Is the manuscript presented in an intelligible fashion and written in standard English?

Reviewer #1: Yes

Reviewer #2: Yes

5. Review Comments to the Author

Reviewer #1: A secondary analysis of the DPP trial which included 317 women with gestational diabetes.

It is an intriuging study but with some caveats: First the risk model is practical and simple and may be used on a primary conselling setting. the simplicity is maybe also the piont of where exact values of risk are not to be trusted: It was a sample of study not geared for pregnancy and the dioagnosis of GDM is based on self-reported statements on what was told during pregnancy. Knowing that glucosuria is a copmmon recurrence in normal pregnancy , too, some misclassification could be present. For second the relative few women to make a model on 5 variables seem to stretch the creditability. Unless, of course, the correlations are that strong that even in each substrata varibales exerts its effect on the outcome. A hint that the sample of 317 may be too skewed is the non-significance of ethnicity. I think too many caucasians with certain traits have been included in the DPP and, consequently, fewer of the ethnic mixed with other diabetogenic backgrounds. Nevertheless, it may be true and work under these circumstances!

This was a secondary analysis of 317 women with a history of GDM and current prediabetes in the DPP study. These women were not pregnant at the time. We have addressed potential for misclassifications in our discussion (lines 414-426) and in item 7 below. In our discussion, we note that the diagnosis of GDM was based on self-reporting; however, studies show women recall their GDM diagnosis and treatment accurately. We agree with the reviewer that the lack of significance of ethnicity, a known risk factor, may have been due to our sample size and race/ethnicity distribution.

Smaller issues:

They introduce the subject by claiming equipotency of metformin and ILI in that both have 50 % reduction diabetes risk; well may be, if applied on similar time shortly before diabetes would manifest itself: while LIL may reduce the risk anytime, metformin is most potent right before the beta-cell goes dry and diabetes would manifest itself

We agree that timing of intervention (i.e., extent of remaining beta cell function) is important in determining relative treatment efficacy. We clarify that treatment response differs based on individual clinical factors (such as remaining beta cell function) in line 52.

titel page: the affilations denoted as superscripts on the authors names (1-5) do not correspond to the institutions (a-d)

We have corrected the affiliations to match with the superscripts.

Reviewer #2: This paper presents important information that adds to the GDM knowledge pool, especially in the risk prediction model of subsequent prediabetes or diabetes. The paper is well written, with a clear text, easy to read. However, I have few comments sated as follows:

1. Method section of the abstract: Though, developed and validated a 3-year diabetes prediction model using Cox proportional hazards regression acceptable but the risk prediction model using 11 baseline clinical variables independently and/ or in combined for risk of prediabetes would be computed by the area under the ROC curve (discriminative power). In addition, a risk score calculation based on a combination of the most pertinent clinical variables model prediction for prediabetes is highly suggested.

Your suggestion is appreciated. We have computed the survival ROC curves for our full and parsimonious models (supplementary Figure 3) and provided the AUC for each model as recommended. 

2. As insulin also given for GDM, why the prediction model is not included about insulin management or why authors focused only lifestyle intervention vs metformin treatment?

The women in the DPP were assigned to ILI or metformin – with participation after a pregnancy, to prevent future diabetes. Both are considered clinical treatment options for diabetes prevention or prediabetes treatment (not insulin). If there is interest in insulin given during GDM to influence future development of diabetes, that information (insulin given during GDM pregnancy) was not collected in the DPP study.

3. There is controversies in some statement in the use of prediabetes vs diabetes eg. the title A Clinical Diabetes Risk Prediction Model For Prediabetic Women With Prior Gestational Diabetes. Meaning that the main outcome focused on “prediabetes”, However the result section of the abstract “ Within three years, 82 (25.9%) women developed diabetes” and Conclusion section of the abstract also “clinical prediction model was developed for individualized decision making for prediabetes treatment in women with prior GDM” . As Prediabtes and Diabetes has different definition and cutoff point of blood glucose levels, the use of the two terms should be managed thorough out the entire document.

We have reviewed our use of “prediabetes” and “diabetes” throughout the manuscript. For our title, prediabetic is used to describe the population for which the model was developed (in addition to the history of GDM). The model was developed for women who had gestational diabetes during a pregnancy and now have prediabetes. Thus, these women have a high risk for developing diabetes. For women who had GDM and now have prediabetes, ADA guidelines recommend both metformin and ILI as “prediabetes treatment” for diabetes prevention.

Reference: https://care.diabetesjournals.org/content/43/Supplement_1/S32

4. The introduction section page 6 line 91-105: What is the relevance of the stated sentences about model comparison here? I suggested to remove this and described in detail in the method section.

We appreciate your suggestion. In our introduction, we wanted to acknowledge the other models that have been published for predicting diabetes in women with a history of GDM and their limitations for clinical use. 

5. Page 7 line 111 “ ------ neither (placebo)” and result section page 13 Table 1 “Placebo (N=107)” How could the GDM patient in placebo. They should get at least the minimum standard treatment. It has also the ethical concerns?? Clarification is needed

For clarification, these women had prior GDM, and upon enrollment had prediabetes. At the time, the study was conducted 20 years ago, there was uncertainty of ILI and metformin vs. placebo in terms of effective diabetes prevention for these women. Thus, at the time, it was considered clinical equipoise. Standard lifestyle recommendations were provided to all participants randomized to metformin or placebo. We have added this detail to the study design description (lines 104-105).

6. Method section Page 7 line 120 “ --- placebo on the development of diabetes over an average of 3.2 years” Confusing ???

We have changed the statement to “The DPP was a randomized, controlled clinical trial comparing the effectiveness of ILI, metformin (850 mg twice daily), and placebo for diabetes prevention over a mean follow-up period of 3.2 years.” (lines 103-106)

7. Page 8 line 134-136:“Women who answered the question, “Have you ever been told that you had a high sugar level or that you have diabetes” and selected the answer “Only during pregnancy” were considered to have had GDM”. What if there is undiagnosed preexisting diabetes mellitus?

As you point out, it is possible that some women who are diagnosed with GDM may, in fact, have undiagnosed pre-existing diabetes. Women with pregestational diabetes will likely continue to have diabetes on postpartum screening and thereafter; therefore they would not have qualified for the DPP study. At the time of the DPP, there was limited use of first trimester screening compared to now. More likely, an answer with “Only during pregnancy” underestimates the number of women with a prior history of GDM. Women with a history of GDM and prediabetes (prediabetes was a DPP inclusion criteria and diabetes was an exclusion criteria) may not have checked “only during pregnancy” because they may have been diagnosed with glucose intolerance after their GDM pregnancy and therefore we may be underestimating women with prior GDM. We note these potential limitations in our discussion (lines 420-426).

8. As authors stated “ The outcome measure was the development of diabetes as defined by ADA criteria.( Clinical assessments were performed routinely every six months during the average three years of monitoring. Diagnostic criteria for diabetes was defined by a fasting plasma glucose ≥ 140 mg/dL (until June 23, 1997) or ≥ 126 mg/dL (on or after June 24, 1997), or a 2-hour 75-gram oral glucose tolerance test (OGTT) ≥ 200 mg/dL.(20) The diagnosis of diabetes was confirmed if consecutive testing with the same criteria, usually within 6 weeks, was met.” The Outcome Measures ascertainment is confusing eg. FPG ≥ 140 mg/dL or ≥ 126 mg/dL ???? This leads missclassifications bias. Clarification is needed

The potential for misclassification was our concern too. In the methods, we clarify that the change in diagnostic criteria during the study is based on a change to the American Diabetes Association (ADA) diagnostic criteria in 1997. However, this change in diagnostic criteria did not ultimately affect the outcome and have added details to the methodology (lines 158-160, 168-175).

9. Method section: ROC analysis for “Prediction risk model development and evaluation” and its main findings eg. AUC in the result section and supported by figures also suggested.

Our response to item #1 addresses this.

10. Adding the implication of main findings has recommend to make stronger discussion section

We currently describe some of the implications in the discussion:

Risk prediction models can facilitate medical decision making, and may be more accurate or, at least, less biased than subjective predictions. An evidence-based GDM risk estimate can provide individualized health information to women and clinicians to guide prediabetes treatment discussion. The model equation (now in Table 4) may help clinicians estimate the risk of progression to diabetes and magnitude of risk reduction with different prediabetes treatment options (i.e., metformin and/or ILI). To realize the benefits of risk prediction, clinical implementation of decision-making activities including aids must be further investigated.

Since the manuscript was submitted, we have piloted a decision aid that targets with women with prior GDM. We added the following sentence to the discussion (lines 405-409) : “We have developed a decision aid targeting women with prior GDM to assist in decisions regarding diabetes prevention therapy. Models such as those derived from the DPP may be incorporated into decision aids to provide users with calculated diabetes risk scores, including with ILI, metformin, or no therapy.” 

11. Discussion section. Page 21 line 252-264. The paragraphs is not related with your findings eg. Plasma lipidomic analysis, common genetic loci…

This paragraph has been removed.

6. PLOS authors have the option to publish the peer review history of their article (what does this mean?). If published, this will include your full peer review and any attached files.

We have answered “Yes”

Do you want your identity to be public for this peer review? For information about this choice, including consent withdrawal, please see our Privacy Policy.

Reviewer #1: Yes: Finn Lauszus

Reviewer #2: Yes: Achenef Asmamaw Muche

---

## [Decision Letter · Decision Letter 1]

9 May 2021

PONE-D-20-34391R1

A Clinical Diabetes Risk Prediction Model For Prediabetic Women With Prior Gestational Diabetes

PLOS ONE

Dear Dr. Man,

Thank you for submitting your manuscript to PLOS ONE. After careful consideration, we feel that it has merit but does not fully meet PLOS ONE’s publication criteria as it currently stands. Therefore, we invite you to submit a revised version of the manuscript that addresses the points raised during the review process.

We look forward to receiving your revised manuscript.

Kind regards,

Andreas Beyerlein

Academic Editor

PLOS ONE

Journal Requirements:

Additional Editor Comments (if provided):

The authors improved their manuscript considerably and responded well to my comments. I am willing to accept it for publication when they will put their complete analysis code together with a data dictionary into a publicly available online repository and mention the respective URL in the Methods section.

Reviewers' comments:

Reviewer's Responses to Questions

**Comments to the Author**

1. If the authors have adequately addressed your comments raised in a previous round of review and you feel that this manuscript is now acceptable for publication, you may indicate that here to bypass the “Comments to the Author” section, enter your conflict of interest statement in the “Confidential to Editor” section, and submit your "Accept" recommendation.

Reviewer #2: All comments have been addressed

2. Is the manuscript technically sound, and do the data support the conclusions?

Reviewer #2: Yes

3. Has the statistical analysis been performed appropriately and rigorously? 

Reviewer #2: Yes

4. Have the authors made all data underlying the findings in their manuscript fully available?

Reviewer #2: (No Response)

5. Is the manuscript presented in an intelligible fashion and written in standard English?

Reviewer #2: Yes

6. Review Comments to the Author

Reviewer #2: (No Response)

7. PLOS authors have the option to publish the peer review history of their article (what does this mean?). If published, this will include your full peer review and any attached files.

Reviewer #2: No

---

## [Author Response · Author response to Decision Letter 1]

14 May 2021

We have provided the URL to the analysis code in the methods section.

---

## [Editor Report · Decision Letter 2]

18 May 2021

A Clinical Diabetes Risk Prediction Model For Prediabetic Women With Prior Gestational Diabetes

PONE-D-20-34391R2

Dear Dr. Man,

We’re pleased to inform you that your manuscript has been judged scientifically suitable for publication and will be formally accepted for publication once it meets all outstanding technical requirements.

Kind regards,

Andreas Beyerlein

Academic Editor

PLOS ONE
---

## [Editor Report · Acceptance letter]

17 Jun 2021

PONE-D-20-34391R2 

A Clinical Diabetes Risk Prediction Model For Prediabetic Women With Prior Gestational Diabetes 

Dear Dr. Man:

I'm pleased to inform you that your manuscript has been deemed suitable for publication in PLOS ONE. Congratulations! Your manuscript is now with our production department. 

Kind regards, 

on behalf of

Dr. Andreas Beyerlein 

Academic Editor

PLOS ONE